# Isolation and Characterization of Novel Bacteriophages to Target Carbapenem-Resistant *Acinetobacter baumannii*

**DOI:** 10.3390/antibiotics13070610

**Published:** 2024-06-29

**Authors:** Yoon-Jung Choi, Shukho Kim, Minsang Shin, Jungmin Kim

**Affiliations:** Department of Microbiology, School of Medicine, Kyungpook National University, Daegu 41944, Republic of Korea; yjchoi8727@knu.ac.kr (Y.-J.C.); shukhokim@knu.ac.kr (S.K.); shinms@knu.ac.kr (M.S.)

**Keywords:** bacteriophage, phage therapy, carbapenem-resistant *Acinetobacter baumannii* (CRAB)

## Abstract

The spread of multidrug-resistant *Acinetobacter baumannii* in hospitals and nursing homes poses serious healthcare challenges. Therefore, we aimed to isolate and characterize lytic bacteriophages targeting carbapenem-resistant *Acinetobacter baumannii* (CRAB). Of the 21 isolated *A. baumannii* phages, 11 exhibited potent lytic activities against clinical isolates of CRAB. Based on host spectrum and RAPD-PCR results, 11 phages were categorized into four groups. Three phages (vB_AbaP_W8, vB_AbaSi_W9, and vB_AbaSt_W16) were further characterized owing to their antibacterial efficacy, morphology, and whole-genome sequence and were found to lyse 37.93%, 89.66%, and 37.93%, respectively, of the 29 tested CRAB isolates. The lytic spectrum of phages varied depending on the multilocus sequence type (MLST) of the CRAB isolates. The three phages contained linear double-stranded DNA genomes, with sizes of 41,326–166,741 bp and GC contents of 34.4–35.6%. Genome-wide phylogenetic analysis and single gene-based tree construction revealed no correlation among the three phages. Moreover, no genes were associated with lysogeny, antibiotic resistance, or bacterial toxins. Therefore, the three novel phages represent potential candidates for phage therapy against CRAB infections.

## 1. Introduction

*Acinetobacter baumannii*, belonging to the *Moraxellaceae* family, is notorious for its persistence in hospital settings and rapid evolution of resistance to various antibiotic classes [1,2,3,4]. Although colistin (polymyxin) is considered the last line of defense against multidrug-resistant Gram-negative bacterial infections, carbapenem-resistant *A. baumannii* (CRAB) infections are increasing worldwide [3,4,5]. Thus, mortality and healthcare costs are growing owing to prolonged hospitalization [3,6,7,8].

Multilocus sequence typing (MLST) is a precise method of characterizing bacterial isolates based on several housekeeping gene sequences [9,10,11]. MLST is useful for tracking the epidemiology of infections, understanding bacterial population structures, and observing the emergence and spread of antibiotic resistance [11,12]. Additionally, MLST is used to identify and differentiate between strains of *A. baumannii*, which is particularly useful for epidemiological tracking and infection control purposes [9,10]. Indeed, various MLST schemes have been developed for *A. baumannii*, although the Pasteur Institute and Oxford schemes are the most popular [11,12]. The *A. baumannii* MLST scheme employs fragments from seven housekeeping genes: *cpn60* (60-KDa chaperonin), *fusA* (elongation factor EF-G), *gltA* (citrate synthase), *pyrG* (CTP synthase), *recA* (homologous recombination factor), *rplB* (50s ribosomal protein L2), and *rpoB* (RNA polymerase subunit B) [13]. Due to the high genetic variability and adaptability of *A. baumannii*, many sequence types (STs) have been reported. Specific STs are associated with outbreaks and may be more common in hospital-acquired infections. ST191, ST208, ST369, and ST451 are clinical isolates of *A. baumannii* found in South Korea [6,14,15]. These STs are associated with multidrug resistance (MDR) [16].

Bacteriophages (phages) are viruses that infect bacteria, and phage therapy was first used in humans in 1919 [17,18,19,20,21,22]. Since then, phage therapy has received significant attention as a complementary or alternative approach to traditional antibiotic therapies. Phages selectively infect and lyse bacteria, exerting a minimal effect on the microbial community while eliminating pathogenic bacteria [22]. Moreover, the high specificity of phages, their ability to self-replicate at the infection site, and coevolutionary potential with bacteria are considered significant advantages of phage therapy. Recently, Li et al. reviewed the progress in isolating, genome sequencing, preclinical research, and clinical application of *A. baumannii* phages [23]. They summarized the characteristics of 132 *A. baumannii* phages sequenced and investigated until October 2022. According to their report, most *A. baumannii* phages contained linear double-stranded DNA, and their genome size was 4–234 kb. They also reported that the 132 *A. baumannii* phages exclusively belonged to clades within the order *Caudovirales*, consisting of *Myovirus* and *Podovirus* [24,25].

In this study, we isolated new phages capable of lysing CRAB clinical isolates and explored their potential as candidates for phage therapy. We also investigated their lytic spectra against CRAB clinical isolates of different STs.

## 2. Results

### 2.1. Bacteriophage Isolation, Host Spectrum Analysis, and RAPD–PCR Classification

We isolated 21 phages from the 20 hospital sewage samples collected between April and October 2022 using the *A. baumanni* standard strain ATCC17978. Among the 21 phages, 11 demonstrated potent lytic activities against CRAB clinical isolates. Spot tests were conducted on 29 clinical strains of various CRAB STs, and the 11 phages exhibited different lytic activities according to the CRAB STs (Table 1). Phage AW8 demonstrated clear lysis (CL: clear transparent lysis zone without any resistant colony) capability against seven CRAB strains belonging to ST 229 and ST 784 and opaque lysis (OL: non-transparent, opaque lytic zone) ability against some strains belonging to ST208, ST357, and ST369. Phage AW9 exhibited lytic capabilities against various CRAB STs; however, it demonstrated only CL ability against CRAB belonging to ST784; semi-clear lysis (SCL: semi-clear lysis zone with some resistant colony) ability against CRAB belonging to ST357, ST369, and ST451; a mixture of SCL and OL ability against CRAB belonging to ST191 and ST208; OL ability against ST229 strains. Phages isolated from the same sewage sample (AW15-1–15-4, AW16-1–16-5) exhibited similar phagograms, indicating their possible similarity. AW15-1–15-4 and AW16-1 demonstrated CL ability against ST229, ST552, and ST784 and two types of lytic abilities (CL and SCL) against ST357. AW16-2–AW16-4, unlike AW16-1, exhibited OL lytic ability against *A. baumannii* ATCC 19606 (Table 1).

The efficiency of plating (EOP) was used to define the effectiveness of bacteriophage against target bacteria. EOP values of 0.1–1.0 were ranked as high efficiency. In contrast, an EOP value of 0.001 to 0.99 was categorized as low efficiency. Three isolated phages showed activity only in infecting specific bacteria in their bacterial host (Appendix A). Phage AW8 showed high efficiency with ST191 and ST784, and phage AW16 demonstrated high efficiency with ST552, ST229, ST357, and ST784. Phage AW9 only showed high efficiency with ST784 and exhibited low EOP with ST357, ST191, ST369, ST451, and some ST208 (LIS20132370 and KBN10P04322).

Based on RAPD–PCR results, 11 phages were classified into four clear clusters (Appendix A). Phages AW15-1–15-4, AW16-1, and AW16-3 through 16-5 were presumed to be similar in the host spectrum and EOP, each demonstrating the same RAPD–PCR patterns. Out of the 11 phages, three from different classes, AW8, AW9, and AW16, were selected to investigate their phenotypic and genotypic characteristics (Table 2).

### 2.2. Phenotypic and Genotypic Characteristics of Three Novel Phages

The plaque morphology of the three phages was quite different from each other, as shown in Table 2. Phage AW8 showed transparent, round plaques with 0.5–0.7 cm diameters, surrounded by halos (clearance zones around each plaque). These plaques developed after 18 h of incubation at 37 °C and presented a tendency to grow larger over time. Phage AW9 had smaller plaque sizes than other phages (0.1–0.2 cm); its lytic activity was relatively weak, and small opaque bacterial colonies appeared in the center of the plaques. Phage AW16-4 morphologies produce highly lytic plaques with 0.2–0.3 cm diameters and reduced lytic activities.

TEM examination of the phages also revealed the unique characteristics of each isolated phage (Table 2). Phage AW8 showed a capsid diameter of 68.54 ± 3.61 nm (*n* = 10), a tail length of 2.75 ± 2.74 nm (*n* = 3), and a width of 36 ± 2 nm. Its features were consistent with those of the *Podovirus* family. Phage AW9 had a capsid diameter of 40.10 ± 2.65 nm (*n* = 15) and a tail measuring 145.45 ± 4.17 nm with a width of 5.68 ± 4.44 nm (both *n* = 15). Phage AW16-4 had a capsid diameter of 38.2 ± 1.75 nm, a tail length of 112.5 ± 4.80 nm, and a width of 5.6 ± 2.34 nm (all *n* = 10). The TEM data revealed that the morphologies of phages AW9 and AW16-4 align closely with the characteristics of the *Myovirus* family.

As listed in Table 2, phages AW8 and AW9 exhibited rapid adsorption to bacteria, with 90% adsorbed within just 1 min. Phage AW16-4 reached a 90% adsorption rate within 3 min of infection. Phage AW8 had a latent period of 10 min, and the burst size was 164 PFU/cell. Phage AW9 had a latent period of 20 min and a burst size of 117 PFU/cell. Phage AW16-4 had a short latent period of 5 min, and its burst size was high at 1012 PFU/cell.

Whole-genome sequencing was conducted on the four phages, AW8, AW9, AW15-2, and AW16-4, confirming that AW15-2 and AW16-4 had identical genome sequences. We designated the phages AW8, AW9, and AW16-4 as *Acinetobacter* phage vB_AbaP_W8 (PP174318), *Acinetobacter* phage vB_AbaSi_W9 (PP146379), and *Acinetobacter* phage vB_AbaSt_W16 (PP174317), respectively (Table 3).

Phage vB_AbaP_W8 revealed a genome size of 41,326 bp and a GC content of 39.2% (Table 3). This phage demonstrated a 94.73% similarity (with a query cover of 90%) to the *Acinetobacter* phage vB_AbaP_B09_Aci08, whereas vB_AbaP_AW8 and vB_AbaP_B09_Aci08 demonstrated a pairwise identity of 45.64% based on MAFFT analysis (Table 3, Figure 1a and Figure 2a). Phage vB_AbaSi_W9 was slightly larger, with a genome size of 43,022 bp and a GC content of 45.6% (Table 3). It demonstrated 96.62% similarity (query cover: 99%) with *Acinetobacter* phage IMEAB3 (Table 3, Figure 1b and Figure 2b). Based on MAFFT analysis, vB_AbaSi_W9 exhibited a pairwise identity of 83.1% with IMEAB3 and 24.7% with Loki, whereas IMEAB3 and Loki shared a pairwise identity of 62.7%. Phage vB_AbaSt_AW16, with a genome size of 166,741 bp and a GC content of 34.4%, demonstrated both a 98.83% (query cover: 94%) similarity to *Acinetobacter* phage ZZ1 and a 66.3% pairwise identity with the same phage based on the MAFFT analysis (Table 3, Figure 1c and Figure 2c).

The predicted ORFs were classified into early, middle, late, and hypothetical proteins (Table 3 and Figure 1). In the early category, genes related to nucleotide metabolism involved in translation and transcription were classified. The middle category included genes related to structure and packaging, while the late category included genes related to lysis, such as endolysin and holin. Phage vB_AbaP_W8 could encode 48 ORFs, including 23 functional and 25 hypothetical proteins. Of these, 11 ORFs were in the early category, and 10 ORFs in the middle category (Table 3 and Figure 1a). Phage vB_AbaSi_W9 had 56 ORFs, including 27 hypothetical proteins, 12 in the early and 15 in the middle categories (Table 3 and Figure 1b). Phage vB_AbaSt_W16 found 112 functional proteins among 242 functional genes, with 85 genes related to nucleotide metabolism in the early category and 25 genes related to structure and packaging in the middle category (Table 3 and Figure 1c). All three phages also had genes for endolysin and holin in the late category (Figure 1c). None of the three phages (vB_AbaP_W8, vB_AbaSi_W9, and vB_AbaSt_W16) were found to contain genes related to pathogenicity/toxicity or lysogenicity.Proteome-based phylogenetic relationships among bacteriophages were discerned using VipTree, whereas relationships through whole-genome similarities were explored using VICTOR Tree. Additionally, we generated a proteome-based phylogenetic tree for taxonomic classification using VipTree, which indicated that although the three phages exhibited similarities to other phages that infect members of Pseudomonadota, they were not phylogenetically related to one another (Table 2 and Figure 3). According to our findings, vB_AbaP_W8 was categorized under the *Autographiviridae* family, vB_AbaSi_W9 remained unclassified, and vB_AbaSt_W16 belonged to the *Straboviridae* family (Figure 3a). Phages vB_AbaP_W8, vB_AbaSi_AW9, and vB_AbaSt_AW16 were designated as *Friunavirus*, *Lokivirus*, and *Zedzedvirus*, respectively, in both the phylogenetic analysis and single gene-based trees, with each being identified as a distinct species (Table 2 and Figure 3b).

## 3. Discussion

We successfully isolated and characterized new bacteriophages with lytic activity against CRAB clinical isolates. We isolated 21 *A. baumannii*-specific phages and elucidated the characteristics of 11 phages that demonstrated potent lytic activity against CRAB clinical isolates. After evaluating the host range, plaque morphology, TEM data, and RAPD–PCR and WGS analysis results, we ultimately identified the 11 phages as three distinct types. These three types were named *Acinetobacter* phage vB_AbaSi_W8, *Acinetobacter* phage vB_AbaSi_W9, and *Acinetobacter* phage vB_AbaSi_W16, respectively.

The three new phages exhibited variable lytic activities against various CRAB STs used in this study, with one of the phages identified as able to lyse 29 CRAB strains belonging to eight STs. Phage vB_AbaP_W8 demonstrated strong lytic activity against ST229 and ST784. Phage vB_AbaSi_W9 exhibited lytic activity against ST191, ST208, ST357, ST369, ST451, and ST784. Phage vB_AbaSt_W16 showed strong lytic activity against ST229, ST357, ST552, and ST784. However, vB_AbaSi_W8 and vB_AbaSi_W16 showed a narrower lysis spectrum against CRAB strains than vB_AbaSi_W9; they accounted for 24.14%–34.48% of the strains demonstrating clear lysis (CL) in spot tests. Conversely, although vB_AbaSi_W9 exhibited a relatively wide lysis spectrum, CL was observed in 10.34% of cases, and the presence of SCL and OL activities suggested that the lytic intensity was relatively weak. In the plaque assay, phages that exhibited CL activity in the spot tests were amplified to >10^5^ PFU/mL, whereas those that exhibited SCL activity were amplified to <10^5^ PFU/mL and did not increase in number even with several additional amplification attempts. Phages that exhibited OL activity did not form plaques in the plaque assay; thus, it is presumed that the OL effect demonstrated in the spot tests was due to enzymes such as polymerases or lysozymes generated from the phages. Proteins such as polymerases, lysozymes, and holins were confirmed in the genomes of all three isolated phages [21]. To determine the host range for the ESKAPE strains, which poses challenges in antibiotic resistance alongside *A. baumannii*, we performed a spot analysis on *E. coli*, *S. aureus*, *K. pneumoniae*, *P. aeruginosa*, *E. faecalis*, and *E. faecium*. The three phages (vB_AbaP_W8, vB_AbaSi_W9, and vB_AbaSt_W16) demonstrated no activity against these strains. Based on the TEM analysis, vB_AbaP_AW8 was classified as belonging to the *Podovirus* family, whereas vB_AbaSi_AW9 and vB_AbaSt_AW16 were classified as members of the *Myovirus* family. However, based on the complete genome sequence analysis, the three phages were classified as *Friunavirus* (vB_AbaP_W8), *Lokivirus* (vB_AbaSi_W9), and *Zedzedvirus* (vB_AbaSt_W16). All three phages demonstrated no virulence, antibiotic resistance, or bacterial toxin-related genes in the WGS analysis. Phages with such characteristics are considered to have minimal potential for contributing toward antibiotic resistance dissemination, rendering them suitable for use in treating bacterial infections.

Determining the host range demonstrated that these three phages could lyse a significant proportion of the tested CRAB strains, and their lytic activity was dependent on the MLST of *A. baumannii* [4,26,27,28]. ST191 was the most prevalent in the International Clonal Lineage II (ICLII) [6,14,15]. The emerging clones ST784 and ST451 fell within this lineage [15]. *A. baumannii* ST191 was classified as a high-risk clone with a mortality rate of > 60%. In contrast, the single gene variant of *gpi* in ST451 and ST784 spread widely and should be closely monitored [5,14,26]. Therefore, the three isolated phages in this study may have potential therapeutic agents against important CRAB strains.

Essoh et al. (2019) investigated phages analogous to phage vB_AbaP_W8 and identified five distinct phages from sewage waters in Abidjan, Ivory Coast [29]. Remarkably, vB_AbaP_46-62_Aci07 and vB_AbaP_B09_Aci08 emerged as close relatives to vB_AbaP_W8, both belonging to the *Podovirus* family, specifically the *Frilvirus* genus. Their genome sizes were 42,330 and 42,067 bp, respectively, slightly larger than the 41,326 bp genome of vB_AbaP_W8. Although their host range and interactions with different STs remain unexplored, morphologically, both phages resembled phage vB_AbaP_W8, with TEM imaging revealing further structural similarities. Nevertheless, at 60 ± 2 nm, their capsid size was approximately 8 nm smaller than that of phage vB_AbaP_W8 [29].

In contrast to the *Acinetobacter* phage IMEAB3 from China, the UK-isolated vB_AbaS_Loki phage exhibited similarities to IMEAB3 [30,31]. Moreover, vB_AbaSi_W9, with a pairwise identity of 96.6%, showed a similarity to IMEAB3, although it shared only a 24.7% pairwise identity with *Acinetobacter* phage vB_AbaS_Loki. Phage vB_AbaS_Loki demonstrated a broad host spectrum, uniquely targeting *A. baumannii* ATCC 17978 [31]. In a fascinating parallel, vB_AbaS_W9, despite being isolated from the clinical strain LIS20130567, optimally amplified in *A. baumannii* ATCC 17978, echoing the specificity of vB_AbaS_Loki. Both phages exhibited EOP (plating efficiency) exclusively with *A. baumannii* ATCC 17978 [31]. Furthermore, although all EOPs for vB_AbaSi_W9 were confirmed when tested with other clinical strains labeled as SCL, the phage titer remained capped at 10^5^ PFU. Morphologically, although vB_AbaSi_W9 is similar to the *Myovirus* family members, vB_AbaS_Loki exhibited characteristics similar to those of the *Sipovirus* family [31].

Compared with the previously investigated *Acinetobacter* phage ZZ1 isolated from Guangzhou and the isolated phage vB_AbaSt_AW16, we observed a genetic similarity of 66.3% pairwise identity through MAFFT analysis [23,32,33]. However, there was a significant distinction in their genetic compositions. Specifically, the early proteins in phage vB_AbaSt_AW16, such as the inhibitor of host Lon protease, polynucleotide kinase, and RIII, did not correspond with the ZZ1 coding regions for middle-associated structural proteins, such as the tail protein and tail fiber protein. Remarkably, the alignment of these nonmatching regions exhibited <95% similarity, underscoring substantial genetic variations between the two phages. Moreover, although ZZ1 exhibited lytic activity against only 3 of 23 antibiotic-resistant *A. baumannii* strains without categorizing them by STs, this divergence suggests potential differences in their life cycles, infectivity, and evolutionary pathways [32,33].

In this study, we explored the unique host ranges of bacteriophages, particularly focusing on their interactions with the varied STs of *A. baumannii*. We aimed to comprehensively map these interactions, shedding light on the intricate relationships between individual phages and their respective bacterial STs. The complex dynamics of bacteria–phage interactions are crucial for developing effective phage therapies. Recent studies have highlighted the coevolutionary arms race between bacteria and phages, emphasizing the need to understand these interactions to harness phages as viable alternatives to antibiotics [34]. Thus, we anticipate that our findings will enhance the current understanding of bacteriophage–host dynamics and provide a foundation for developing tailored phage therapies specifically designed for CRAB infections. Future research must focus on several important aspects. First, investigating the specific interactions and host lytic mechanisms between *A. baumannii* and phages will deepen our understanding of phage biology and facilitate the development of optimized phage-based therapies [21,35,36]. Second, conducting in vivo studies and clinical trials to evaluate the efficacy and safety of phage-based therapies targeting specific STs is crucial [37,38]. This research will provide valuable insights into the real-world therapeutic potential of using these phages and lay the groundwork for their clinical application. Third, investigating the development of phage cocktails that combine multiple phages and exploring combination strategies with antibiotics can enhance therapeutic effectiveness and reduce the risk of developing resistance [39,40,41]. Finally, conducting research to produce antimicrobial proteins using phage-derived enzymes such as endolysin and other polymerases can provide an alternative approach to address the problem of antibiotic resistance [42,43]. When combined with phages, these proteins can work synergistically to combat infections.

## 4. Materials and Methods

### 4.1. Sample Collection and Processing

We collected 20 sewage samples (150 mL per sample) from Kyungpook National University Hospital, Daegu, South Korea, between April and October 2022. The samples were refrigerated at 4 °C and processed within 24 h of collection. They were centrifuged at 7000× *g* for 20 min, filtered through a 0.45 µm filter, and treated with chloroform (final concentration 10% *v*/*v*) to eliminate residual bacteria and release potential phages.

### 4.2. A. baumannii Strains Used in This Study

We obtained 29 CRAB clinical isolates from the Pathogen Resource Bank at Kyungpook National University Hospital from patients hospitalized during 2013–2019. These 29 CRAB strains belonged to the eight types identified during MLST, determined by the Oxford scheme [11,13]. *A. baumannii* standard strains ATCC 17978 and ATCC 19606 were also used for the experiments. The bacteria were cultured on blood agar plates at 37 °C for 24 h and then in the broth of brain heart infusion (BHI, DB, Sparks, MD, USA). Subsequently, they were stored at −70 °C as 15% glycerol stocks.

### 4.3. Isolation and Purification of Bacteriophages

The isolation and purification of bacteriophages were performed using a combination of liquid amplification and agar overlay methods, following established protocols with some modifications [20,44,45,46]. *A. baumannii* was cultured in BHI broth until it reached an optical density (OD_600_) of 0.5, corresponding to a bacterial concentration of approximately 10^8^ CFU/mL. Hospital sewage, filtered as described earlier, was added as a source of phages, and the mixture was incubated initially at 30 °C with 150 rpm agitation and then at 4 °C. Next, the culture was centrifuged, and the supernatant was passed through a 0.45 μm filter. Chloroform was added to a final concentration of 10% to lyse any remaining bacterial cells. Then, the filtrate was mixed with *A. baumannii*, which had been cultured to log phase (OD_600_ = 0.5) in saline magnesium buffer (SM buffer; 50 mM Tris–HCl, 150 mM NaCl, 10 mM MgCl_2_, 2 mM CaCl_2_, pH 7.5). This mixture was combined with 0.75% BHI soft agar and poured onto a BHI agar plate. After overnight incubation at 37 °C, a single plaque was picked and resuspended in 300 μL SM buffer in a sterile 1.5 mL microcentrifuge tube, vortexed, centrifuged, and filtered. Chloroform was added again to a final concentration of 10%. The purified phage suspension was mixed with *A. baumannii* grown to an OD_600_ of 0.5 and incubated at 30 °C and 150 rpm for 16 h. Post incubation, the culture was centrifuged, and the supernatant was filtered. Then, chloroform was added to a final concentration of 10%. This cycle was repeated three times, doubling the volume each time to obtain purified bacteriophages.

To obtain a pure phage solution, we used a modified version of the “Phage on Tap (PoT)” protocol described previously [47]. Further phage purification was conducted using Amicon Ultra-15 Centrifugal filters (10K Merck, Dublin, Ireland). The phage solution obtained through the PoT method was loaded into the filter devices and centrifuged before the filtrate was discarded [47]. Then, the filter devices were refilled with SM buffer and subjected to a second round of centrifugation [47,48]. The remaining solution in the filter devices was collected, mixed with 10% chloroform, and filtered. Finally, the purified phage solution was mixed with glycerol to a final concentration of 15% and stored at −25 °C for future use.

### 4.4. Analysis of Host Spectrum Analysis and Efficiency of Plating (EOP)

To determine the phage host range, a spot titration protocol was used that allowed us to determine both host range and relative phage plating (EOP) [49]. The 29 CRAB isolates and ESKAPE bacteria were used for the host range testing. The ESKAPE bacteria used in this study were *Escherichia coli* (ATCC25922 and KBN10P04004), *Staphylococcus aureus*, *Klebsiella pneumoniae* (ATCC13883 and KBN10P05309), *Pseudomonas aeruginosa* (ATCC27853 and KBN10P01171), *Enterococcus faecalis* (ATCC29212 and KBN10P04153), and *Enterococcus faecium* (KBN10P02059 and KBN10P02311). Bacteria were calibrated to a concentration of 10^8^ CFU/mL, and 200 µL of this sample was added to a 15 mL tube to form a bacterial lawn in 0.75% overlay agar. Phage stocks with a titer exceeding 10^10^ plaque-forming units per milliliter (PFU/mL) were utilized. A volume of 20 µL of this phage stock dilution was then spotted on the bacterial lawn prepared in the overlay agar. The prepared plates were allowed to rest at room temperature for 30 min before incubating at 37 °C overnight. Observable zones of bacterial lawn clearing, indicating cell lysis, were classified as follows: “CL” for clear lysis, “SCL” for semi-clear lysis, “OL” for opaque lysis, “+++” for confluent plaques, and “+/−” for isolated plaques. The presence or absence of halos, defined by reduced turbidity of the bacterial lawn surrounding the plaques or clearings, was also documented. Phage virulence was determined as the efficiency of plating (EOP) [50]. EOP was calculated by dividing the titer of the phage at the terminal dilution on the test strain by the titer of the same phage on its isolation strain. On this basis, phages were classified as highly virulent (1 > EOP > 0.1), moderately virulent (0.001 < EOP < 0.099), avirulent but active (EOP < 0.001), or avirulent (no plaques detected).

### 4.5. Random Amplified Polymorphic DNA–Polymerase Chain Reaction

The genetic diversity of the isolated phages was analyzed by random amplified polymorphic DNA–polymerase chain reaction (RAPD-PCR) [51]. Phage DNA was extracted using the Phage DNA Isolation kit (Norgen, Thorold, Canada) according to the manufacturer’s instructions. The primers used in this study were OPL5 (5′-ACGCAGGCAC-3′), P1 (5′-CCGCAGCCAA-3′), and P2 (5′-AACGGGCAGA-3′), tested at three different concentrations (10 pmol). The PCR mixture containing 20 μL AccuPower^®^ ProFi Taq PCR PreMix (Bioneer, Daejeon, Republic of Korea) and 50 µg of purified phage DNA was subjected to thermal cycling conditions in a Bio-Rad thermocycler (Hercules, CA, USA), which included one cycle at 94 °C for 5 min, followed by 16 cycles of 94 °C for 45 s, 30 °C for 90 s, and 72 °C for 60 s. Another cycle of 94 °C for 3 min was followed by 16 cycles of 94 °C for 30 s, 36 °C for 30 s, and 72 °C for 60 s, and a final step of 75°C for 10 min. The PCR products were visualized on a 1% agarose gel through electrophoresis.

### 4.6. Plaque Formation and Morphology

The phages were plated using the overlay agar plaque assay method described earlier, using the original host for phage isolation [52]. Plates were incubated at 37 °C overnight to facilitate plaque formation. The plaque morphology, size, halo presence, and intensity were analyzed.

### 4.7. Transmission Electron Microscopy (TEM)

Phage morphology was investigated by TEM [49,53,54]. A 20 µL sample of the purified phage suspension, with a 10^10^ PFU/mL titer, was adsorbed onto carbon-coated copper grids and negatively stained with 2% uranyl acetate. The morphology of the phage particles was observed using a TEM instrument (Hitachi HT 7700, Tokyo, Japan).

### 4.8. Adsorption Assay and One-Step Growth Curve Assay

A phage adsorption assay determines the efficiency at which phages attach and infect their bacterial hosts [33,55,56]. This assay helps in understanding the initial phase of the bacteriophage infection process, which is the attachment of the phage to the bacterial cell *A. baumannii* ATCC17978 and KNB10P02782 were cultured in 20 mL of BHI medium until reaching the logarithmic phase (concentration: 1 × 10^8^ CFU/mL) at OD_600_. To this culture, 2 mL of phage solution (concentration: 1 × 10^6^ PFU/mL), filtered using a 0.22 μm syringe filter, was added. The final MOI was 0.001, and the mixture was incubated at 37 °C. Samples of 1 mL were collected at set intervals (0, 1, 2, 3, 5, 10, and 15 min), centrifuged at 13,500× *g* for 3 min, and filtered through a 0.22 μm filter. The plaque assay was performed with the filtrates to determine PFU, calculated by counting 30–300 plaques on the plate when diluted in an SM buffer.

The phages’ latent period and burst size were confirmed through first-stage growth analysis [33,55,56,57]. Host strains of the phages, *A. baumannii* ATCC17978 and KNB10P02782, were grown in 10 mL of BHI broth to a logarithmic growth phase with an OD_600_ of 0.5. After centrifugation at 7000× *g* for 15 min, the resulting bacterial pellet was resuspended in 0.9 mL of prewarmed BHI broth. The phage suspension was introduced into the bacterial mixture at an MOI of 0.01. This mixture was then incubated at 37 °C for 15 min to allow phage adsorption. After this step, the blend was centrifuged at 13,500× *g* for 3 min to remove unattached phages. The now infected bacterial pellet was reconstituted in 10 mL of BHI broth and further incubated at 37 °C. During this process, sampling was conducted at 5 min intervals for up to 60 min. The collected samples were centrifuged at 13,500 rpm for 3 min, and the supernatant was filtered using a 0.22 μm filter. The phage titer of this supernatant was confirmed through the double agar overlay method. Simultaneously, the burst size was calculated by dividing the average PFU/mL of the incubation period by the average PFU/mL of the last three time points of the experiment [33,35,50]. Results are the mean ± standard deviation of three replicates.

### 4.9. Genome Sequencing of Phages and Bioinformatics Analysis

The phage genome was analyzed according to the criteria established by Philipson et al. (2018) for phage genome characterization [58]. The genomic DNA of phages was sequenced using an Illumina Miseq platform (San Diego, CA, USA), and sequencing reads were assembled using the Celemics pipeline (Celemics, Seoul, Republic of Korea). The assembled sequences and open reading frames (ORFs) were predicted using Glimmer 3.02 and the Geneious Prime software 2023 package. Genomic annotation was performed via BLAST searches against the NCBI nonredundant protein database using the Geneious Prime software 2023 package. Utilizing GenBank data, the phage genome maps were constructed using the web-based tools Geneious Prime 2023 and SnapGene 7.2.

The genetic tree was constructed using VICTOR tools (https://victor.dsmz.de, accessed on 12 February 2023) [59]. Comparative genomic analysis was performed using MAUVE 2.4.0 and the Artemis Comparison Tool [60,61]. Comparative genomic and phylogenetic analyses were performed to evaluate the relationship between the isolated phages and available phage sequences in databases. Predictions for virulence factors, antibiotic resistance genes, and transmembrane domains were accomplished using the Virulence Factors Database (VFDB; http://www.mgc.ac.cn/VFs/, accessed on 22 February 2023) and the Comprehensive Antibiotic Research Database (https://card.mcmaster.ca/, accessed on 1 March 2023) [26]. 

### 4.10. Accession Numbers of the Genome Data

The genome sequences of the isolated phages have been deposited in the NCBI database under the accession numbers: vB_AbaP_W8 (PP174318), vB_AbaSi_W9 (PP146379), and vB_AbaSt_W16 (PP174317). The sequencing data can be accessed through these accession numbers for further reference and analysis.

### 4.11. Statistical Analysis

Data were analyzed using one-way analysis of variance (ANOVA). *p*-values lower than 0.05 were considered statistically significant.

## 5. Conclusions

The isolation and characterization of phages specific to CRAB strains represent a significant advancement in phage therapy against antibiotic-resistant *A. baumannii* infections. Therefore, the newly isolated three phages in this study demonstrate promise as potential candidates for phage therapy against CRAB infections.

## Figures and Tables

**Figure 1 antibiotics-13-00610-f001:**
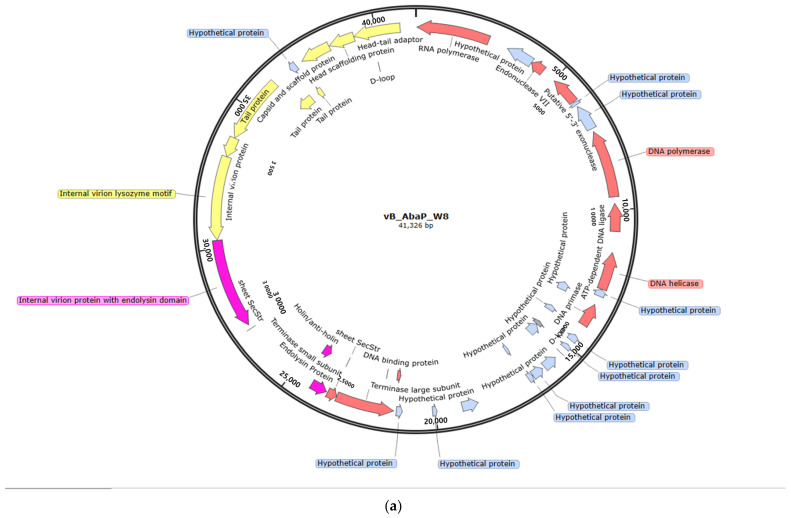
Annotation and map of characterized proteins from phages (**a**–**c**). The various identified modules were based on predicted functions using bioinformatic analysis. Glimmer 3.02 and the Geneious Prime software 2023 package predicted the assembled sequences and open reading frames (ORFs). Genomic map were implemented using SnapGene. The annotations are color-coded as follows: pink represents early proteins involved in translation and transcription, yellow indicates middle genes associated with structural functions, purple denotes late proteins such as endolysin and holin, and light purple corresponds to hypothetical proteins.

**Figure 2 antibiotics-13-00610-f002:**
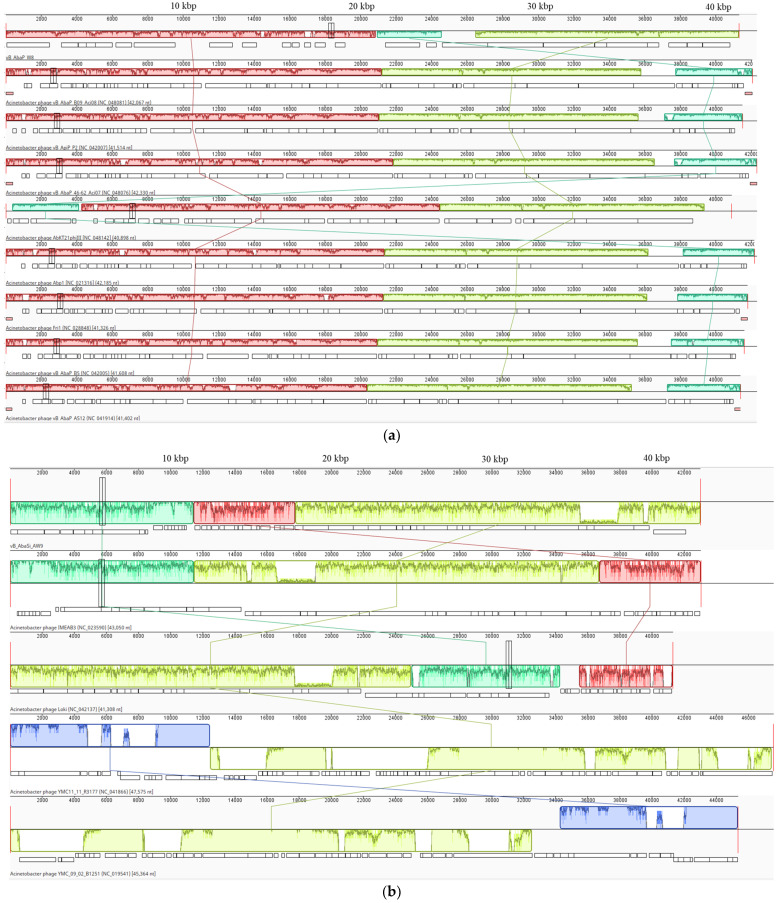
MAUVE software 2.3.1 analysis of phages showing genomic rearrangements (different colors) compared to other phage hits in the NCBI database. Regions highlighted in identical colors across the genomes denote locally collinear blocks. Within each of these blocks, the darker lines illustrate the mean similarity plots along with their respective ranges. White blocks indicate annotated genes with the reverse strand shifted downward. (**a**) Phage vB_AbaP_W8; (**b**) phage vB_AbaSi_W9; (**c**) phage vB_AbaSt_W16.

**Figure 3 antibiotics-13-00610-f003:**
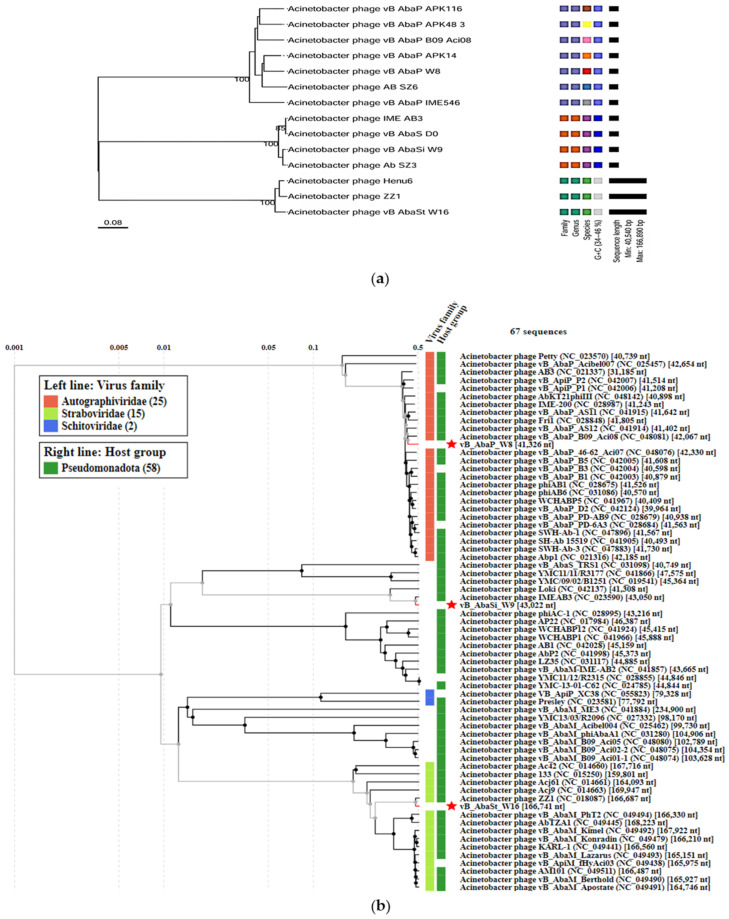
Genomic and phylogenetic comparison analyses of novel phages. (**a**) Phylogenetic analysis of three phages. Bootstrap support (>60) is shown for each node. (**b**) Proteome-based phylogenetic tree of phages constructed by comparison of 64 related taxa using VipTree. Inner and outer rings indicate phage and host taxa at the family level.

**Table 1 antibiotics-13-00610-t001:** Phagogram of 11 isolated bacteriophages against carbapenem-resistant *Acinetobacter baumannii*.

^1^ CRAB	Source	Date Collection	^2^ CCs	^3^ STs	Spot Test Results										
					Φ AW8	Φ AW9	Φ AW15-1	Φ AW15-2	Φ AW15-3	Φ AW15-4	Φ AW16-1	Φ AW16-2	Φ AW16-3	Φ AW16-4	Φ AW16-5
KBN10P02782	Blood	2013	552	552	-	-	CL	CL	CL	CL	CL	CL	CL	CL	CL
LIS20145805	ND	2014	110	229	CL	OL	CL	CL	CL	CL	CL	CL	CL	CL	CL
LIS20145719	ND	2014			CL	-	CL	CL	CL	CL	CL	CL	CL	CL	CL
LIS20144539	ND	2014			CL	OL	CL	CL	CL	CL	CL	CL	CL	CL	CL
KBN10P04598	Blood	2016			CL	-	CL	CL	CL	CL	CL	CL	CL	CL	CL
LIS20130976	Sputum	2013	92	357	-	SCL	CL	CL	CL	CL	CL	CL	CL	CL	CL
LIS20130721	Sputum	2013			-	SCL	CL	CL	CL	CL	CL	CL	CL	CL	CL
LIS20130567	Urine	2013			-	SCL	CL	CL	CL	CL	CL	CL	CL	CL	CL
KBN10P04322	Blood	2016			OL	SCL	SCL	SCL	SCL	SCL	SCL	SCL	SCL	SCL	SCL
KBN10P05102	Blood	2016		784	CL	CL	CL	CL	CL	CL	CL	CL	CL	CL	CL
KBN10P04703	Blood	2016			CL	CL	CL	CL	CL	CL	CL	CL	CL	CL	CL
KBN10P04697	Blood	2017			CL	CL	CL	CL	CL	CL	CL	CL	CL	CL	CL
KBN10P02972	Pus	2013		191	-	SCL	-	-	-	-	-	-	-	-	-
KBN10P02901	ND	2013			-	SCL	-	-	-	-	-	-	-	-	-
KBN10P02755	Blood	2013			-	OL	-	-	-	-	-	-	-	-	-
KBN10P04594	Venous blood	2016			-	OL	-	-	-	-	-	-	-	-	-
KBN10P04627	Central blood	2016			-	SCL	-	-	-	-	-	-	-	-	-
KBN10P04948	Blood	2017			-	SCL	-	-	-	-	-	-	-	-	-
KBN10P02768	Blood	2013		208	-	OL	-	-	-	-	-	-	-	-	-
LIS20132370	ND	2013			OL	OL	-	-	-	-	-	-	-	-	-
KBN10P04322	Blood	2016			OL	SCL	-	-	-	-	-	-	-	-	-
LIS20140444	ND	2014		369	OL	SCL	-	-	-	-	-	-	-	-	-
LIS20138989	ND	2013			-	SCL	-	-	-	-	-	-	-	-	-
LIS20137924	Ascites	2013			-	SCL	-	-	-	-	-	-	-	-	-
KBN10P04633	Blood	2016			-	SCL	-	-	-	-	-	-	-	-	-
KBN10P05663	Blood	2018			-	SCL	-	-	-	-	-	-	-	-	-
KBN10P05982	Blood	2018			-	SCL	-	-	-	-	-	-	-	-	-
KBN10P04600	Blood	2016		451	-	SCL	-	-	-	-	-	-	-	-	-
KBN10P05231	Blood	2016			-	SCL	-	-	-	-	-	-	-	-	-
*A. baumannii* ATCC17978					CL	CL	CL	CL	CL	CL	CL	CL	CL	CL	CL
*A. baumannii* ATCC19606					-	SCL	-	-	-	-	-	OL	OL	OL	OL

^1^ CRAB: carbapenem-resistant *Acinetobacter baumannii*. ^2^ CCs: clonal complexes. ^3^ STs: sequence types.

**Table 2 antibiotics-13-00610-t002:** Summary of the phenotypic characterization of *Acinetobacter baumannii* phages.

Properties of *Acinetobacter baumannii* Phages					
Feature			*Acinetobacter* Phage		
			AW8 (vB_AbaP_W8)	AW9 (vB_AbaSi_W9)	AW16-4 (vB_AbaSt_W16)
Phage characteristics					
Source			Hospital sewage		
Isolation date			2022	2022	2022
Isolation strain			*A. baumannii* ATCC17978		*A. baumannii* clinical isolate KBN10P02782
Summary of phage host spectra: no. lysed/no. tested (% lysed)					
^1^ CC552 (*n* = 1)		^2^ EOP > 0.1	0/1 (0%)	0/1 (0%)	1/1 (100%)
		EOP > 0.001	0/1 (0%)	0/1 (0%)	0/1 (0%)
CC110 (*n* = 4)		EOP > 0.1	4/4 (100%)	0/4 (0%)	4/4 (100%)
		EOP > 0.001	0/4 (0%)	0/4 (0%)	0/4 (0%)
CC92 (*n* = 24)		EOP > 0.1	3/24 (13%)	3/24 (13%)	7/24 (29%)
		EOP > 0.001	0/24 (0%)	23/24 (96%)	0/24 (0%)
Total clinical isolates (*n* = 29)		EOP > 0.1	7/29 (24%)	3/29 (10%)	12/29 (41%)
		EOP > 0.001	0/29 (0%)	23/29 (79%)	0/29 (0%)
Plaque morphology			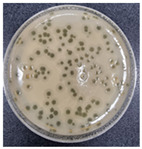	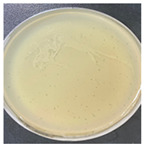	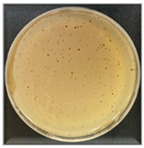
Plaque size (cm)			0.5–0.7	0.1–0.2	0.2–0.3
Stock (PFU/mL)			1.62 × 10^8^	6.75 × 10^14^	2.84 × 10^14^
Transmission electron microscopy (TEM)	Phage morphology		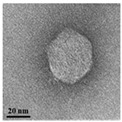	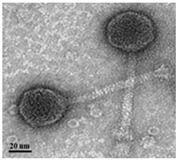	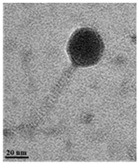
			*Podovirus*	*Myovirus*	
	Tail length		2.75 ± 6.51 nm (*n* = 5)	145.45 ± 4.17 nm (*n* = 15)	112.5 ± 4.80 nm (*n* = 10)
	Tail width		8.74 ± 2.74 nm (*n* = 3)	5.68 ± 4.44 nm (*n* = 15)	5.6 ± 2.34 nm (*n* = 10)
	Capsid dimeter		68.54 ± 3.61 nm (*n* = 10)	40.10 ± 2.65 nm (*n* = 15)	38.2 ± 1.75 nm (*n* = 10)
Growth Properties					
Adsorption constant (PFU/mL)			1 × 10^6^		
90% adsorbed (min)			1	1	3
Latent period (min)			10	20	5
Burst size (PFU/cell)			164	117	1102

^1^ CC: clonal complexes; ^2^ EOP: efficiency of plating.

**Table 3 antibiotics-13-00610-t003:** Summary of the genotypic characterization of *Acinetobacter baumannii* phages.

Attribute		*Acinetobacter* Phage vB_AbaP_W8	*Acinetobacter* Phage vB_AbaSi_W9	*Acinetobacter* Phage vB_AbaSt_W16
Accession no.		PP174318	PP146379	PP174317
Genome size (bp)		41,326	43,022	166,741
GC content (%)		39.2	45.6	34.4
Feature content		48	56	242
Hypothetical proteins		23	27	130
Length of direct terminal repeats (amino acids)		25	29	112
tRNA genes		None	None	10
tRNA genes		None	None	10
Bacterial toxin genes		None	None	None
Antibiotic resistance genes		None	None	None
Genes indicating temperate lifecycle		None	None	None
Family		*Autographiviridae*	Unclassified	*Straboviridae*
Genus		*Friunavirus*	*Lokivirus*	*Zedzedvirus*
Most similar phage sequence	Scientific name	*Acinetobacter* phage vB_AbaP_B09_Aci08	*Acinetobacter* phage IMEAB3	*Acinetobacter* phage ZZ1
	Query cover (%)	90	99	94
	Identity with (%)	94.73	96.62	98.83

## Data Availability

Source data supporting the findings of the present study are included in the article and Appendix A.

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
