# Peer review of "Isolation and Characterization of Novel Bacteriophages to Target Carbapenem-Resistant *Acinetobacter baumannii"

_antibiotics, 2024, doi:10.3390/antibiotics13070610_

Round 1

Reviewer 1 Report

Comments and Suggestions for Authors

The aim of the study entitled “Isolation and characterization of new bacteriophages targeting carbapenem-resistant Acinetobacter baumannii” was to isolate and identify lytic phages for A. baumannii and to explore their potential as candidates for phage therapy.

The methodological design used by the authors allowed the isolation and identification of 3 lytic phages for some STs (sequence types) of A. baumannii. The results were presented in a very clear manner, being the reader able to understand easily that 3 new lytic phages were identify (previously not reported). However, there are some details in the writing of the manuscript that I point out below:

The abbreviations CL, OL and SCL are written for the first time on lines 71 and 74, their meaning must be described.

The numbering of the references in the manuscript is not ordered consecutively. In the introduction it concludes with reference citation no. 25 and continues with no.47 in the discussion section.

Author Response

Comments 1: The abbreviations CL, OL and SCL are written for the first time on lines 71 and 74, their meaning must be described.

Response: We have now provided the full forms and definitions of CL (Clear Lysis), OL (Opaque Lysis), and SCL (Semi-Clear Lysis) upon their first occurrence in the manuscript (lines 74- 79).

Comments 2: The numbering of the references in the manuscript is not ordered consecutively. In the introduction it concludes with reference citation no. 25 and continues with no.47 in the discussion section.

Response: The references have been thoroughly revised and renumbered to ensure they follow a consecutive order throughout the manuscript.

Reviewer 2 Report

Comments and Suggestions for Authors

This is a very interesting study where 3 novel phages have been isolated against carbapenem-resistant Acinetobacter baumannii infections.

Line 28: please note that colistin (polymexin) is the last resort of antibiotic against Gram negative antibiotic – so please edit lines 28 and 29 to provide accurate information and you can add that colistin. 

In the discussion section: (line 291)- authors need to refer to the complex dynamics of bacteria-phage interaction; https://www.ncbi.nlm.nih.gov/pmc/articles/PMC10307801/ and the importance of studying the basis of this interaction in order to ensure developing effective phage therapy as an alternative to antibiotics against bacterial infections. This recent citation  https://www.ncbi.nlm.nih.gov/pmc/articles/PMC10307801/ is important to improve the quality of the manuscript.

In the methods section: authors need to provide a section on the accession numbers of the WGS of isolates phages and where sequencing data is deposited.

Author Response

Comments 1: Line 28: please note that colistin (polymyxin) is the last resort of antibiotic against Gram-negative bacteria – so please edit lines 28 and 29 to provide accurate information and you can add that colistin.

Response: Corrected the information (lines 28-30).

Comments 2: In the discussion section (line 291), authors need to refer to the complex dynamics of bacteria-phage interaction and the importance of studying the basis of this interaction in order to ensure developing effective phage therapy as an alternative to antibiotics against bacterial infections. This recent citation is important to improve the quality of the manuscript: https://www.ncbi.nlm.nih.gov/pmc/articles/PMC10307801/.

Response: We have expanded the discussion section to include the complex dynamics of bacteria-phage interaction, referencing the suggested recent citation to underscore the importance of studying these interactions for effective phage therapy development (lines 246-249).

Comments 3: In the methods section, authors need to provide a section on the accession numbers of the WGS of isolated phages and where sequencing data is deposited.

Response: A section has been added in the methods to provide the accession numbers of the isolated phages' whole genome sequencing (WGS) and indicate where the sequencing data is deposited (lines 392-395).